# Study on the failure of oil-contaminated wheel-rail conditions

**Zhou Gaowei[1,2], Zhou Jiajun[3]*, Tian Chun[3], Fei Gao[1]**

**1** Engineering Research Center of Continuous Extrusion, Ministry of Education, Dalian Jiaotong University, Dalian, China, **2** CRRC Industrial Academy Corporation Limited, Beijing, China, **3** Institute of Rail Transit, Tongji University, Shanghai, China,

* 892149508@tongji.edu.cn

## Abstract

The phenomenon of adhesion improvement during wheel-rail sliding has been experimentally verified under water conditions. However, the academic community is in agreement that, for oil that is also fluid, the adhesion characteristic curve under oil conditions exhibits a single peak, making adhesion improvement through wheel-rail sliding impossible. To investigate whether a similar adhesion improvement phenomenon exists under high-viscosity oil medium conditions as observed under water condition, this study conducted wheel-rail adhesion tests on oil-contaminated interfaces within a slip ratio up to 80%. The test results demonstrate that under higher wheel-rail slip, an adhesion improvement phenomenon also occurs on oil-contaminated rail surfaces, although it is more stringent compared to water conditions. The essence of this adhesion improvement is due to the lubrication failure of oil caused by temperature. Finally, this study analyzes the failure conditions of the oil-contaminated interface and its influencing factors, determining the thermal failure temperature range of the oil film.

## 1. Introduction

The rolling contact between wheel and rail forms a friction pair, the wheel-rail contact system, which is essential for the operation of railway vehicles. This wheel-rail contact is a critical characteristic that unequivocally distinguishes railway vehicles from other non-wheel-rail transportation modes. Railway vehicles rely on this contact to achieve load-bearing, guidance, traction and braking functions. The magnitude of the traction or braking forces between the wheel and rail is directly determined by the tangential forces at the contact patch. However, railways are open systems and are therefore inevitably influenced by the natural environment. The harsh operational conditions and complex open-track environments inevitably lead to the presence of third medium [1] such as water, oil, or leaves on the wheel-rail contact interface, which significantly reduces adhesion coefficient. This directly results in limited braking capacity, extended braking distances, reduced transportation efficiency, and even potential threats to operational safety. As the speed and density of railway vehicle operations continue to increase, the conflict between the growing demand for braking force and low wheel-rail adhesion will become even more pronounced.

**Data availability statement:** All relevant data are within the paper and its Supporting information.

**Funding:** My fund numbers are these two, the CRRC Academy Corporation Limited (No. 2022CYY005) and the National Natural Science Foundation of China (No. 52072266). The funders had no role in study design, data collection and analysis, decision to publish, or preparation of the manuscript.

**Competing interests:** The authors have declared that no competing interests exist.

Traditional theories and experiments suggest that the contact state between wheel and rail remains statically unchanged [2], resulting in a well-defined curve for the adhesion coefficient-slip ratio characteristic. According to this view, the adhesion coefficient increases with the slip ratio, reaches saturation at approximately 1% slip ratio, and then begins to decline. However, an increasing number of scholars now recognize that the condition of the wheel-rail surface is a variable, and that there exists a coupled relationship between wheel sliding behavior and the adhesion coefficient [3]. The wheel's sliding motion can effectively clean the rail surface contaminated by third medium, leading to a secondary rise in the adhesion coefficient as the slip rate increases. This phenomenon is commonly referred to as adhesion improvement or adhesion recovery.

The phenomenon of adhesion improvement under water conditions has been empirically validated by scholars from a range of countries. The earliest field tests can be traced back to the anti-slip tests conducted by the French National Railway Company (SNCF) [4], where Michel Boiteux discovered that after the initial peak observed at low creep, a secondary rise in adhesion and a second peak occurred as the relative sliding speed increased. Similar outcomes were observed in the experiments conducted by Hiraku Tanimoto [5] and Shin-ichi [6] in Japan, where the tangential force in the macroscopic sliding region was found to exceed that in the micro-slip region. Zhou et al. [7,8] conducted tests on the adhesion coefficient under water conditions and identified the phenomenon of adhesion improvement between wheels and rails, with their results demonstrating two distinct forms of adhesion improvement curves. Chang et al. [9–11] conducted adhesion tests in an water-lubricated condition, encompassing both the creep and the subsequent large sliding phases of braking. The findings indicated that the outcomes at lower slip ratio (within 1%) were more aligned with Kalker's theoretical framework. However, during the large sliding phase of braking (with slip ratio up to 20%), a secondary increase in the adhesion coefficient was observed. A series of adhesion tests were conducted by Petr Voltr et al. [12,13] under water conditions using a tram testing platform. The results indicated a trend of increasing adhesion with rising slip ratio. Additionally, Dellmann and Viereck [14], Bosso et al. [15,16], Fontanel et al. [17], Allotta et al. [18,19], and Zhu et al. [20] have conducted research on and employed the phenomenon of adhesion improvement under water conditions. They concluded that under low adhesion conditions, applying a large sliding force to the wheelset and maintaining the sliding duration long enough (approximately several tens of seconds) can result in a curve with a second positive slope and a second peak as the frictional energy is fed back into the wheel-rail contact patch.

However, contaminants on the actual wheel-rail interface extend far beyond just water. Compared to water, oil, a fluid with much higher viscosity, often leads to significantly lower adhesion levels and presents greater safety hazards. Studies by Beagley et al. [21], Kumar et al. [22], Jin et al. [23], Gallardo-Hernandez et al. [24], Lewis et al. [25–31], Arias-Cuevas et al. [32,33], and Wang et al. [34] have investigated the issue of low adhesion under oil-lubricated conditions. By analyzing the effects of factors such as speed, axle load, and oil volume, these studies found that the adhesion coefficient under oil conditions generally drops to 0.05 or lower, reaching an ultra-low adhesion state. Some researchers further measured the oil film thickness and discovered that the film between the wheel and rail is approximately tens of micrometers thick, which is more than ten times thicker than that under water condition [22]. As a result, the oil can lead to an even lower wheel-rail adhesion coefficient. These experimental studies consistently indicate that the adhesion characteristics of the wheel-rail system under oil conditions follow a single-peak curve, with no secondary rise in adhesion observed in the macroscopic slip region. To date, no research has demonstrated that wheel sliding under oil conditions can lead to an adhesion improvement phenomenon similar to that observed under water conditions.

Why does wheel sliding lead to adhesion improvement under water conditions but not under oil conditions, despite both being fluids? The author's analysis of existing literature suggests that significant sliding levels are required for adhesion improvement to occur. Typically, a secondary rise in adhesion is observed at slip ratio of 10–20% for water-contaminated interfaces, while adhesion characteristics under lower slip levels also display a single-peak feature. Most current research on oil-contaminated rail surfaces focuses on adhesion characteristics within slip ratio up to 30%, lacking exploration into higher slip ratio ranges of 30% and above. This limitation has resulted in the oil medium exhibiting a single-peak adhesion characteristic. If the slip ratio between the wheel and rail could be increased further, potentially approaching wheel locking conditions, might the oil-contaminated condition also exhibit adhesion improvement?

To address this issue, this study utilized a circumferential test rig to construct oil-contaminated wheel-rail contact interfaces. Adhesion characteristic tests were conducted over a slip ratio range of up to 80% to determine whether adhesion improvement phenomena can occur with high-sliding under oil conditions.

## 2. Experimental equipment and procedure

### 2.1. Experimental equipment

In this study, a circumferential test rig was employed to investigate adhesion characteristics under oil conditions. The test rig features a track-fixed, arm-driven wheel/rail configuration and operates as a 1:4 scale model with a maximum designed speed of up to 160 km/h, as detailed in Table 1. The mechanical structure of the rig consists of a frame, box cover, rotating arm, wheels, annular rail, fluid spraying device, motor, coupling, clutch, and other internal transmission components. Additionally, the test rig is equipped with a thermal imager for real-time monitoring of the wheel tread temperature. The overall layout of the test rig is illustrated in Fig 1.The test rig uses a circular track structure, which induces a spin moment within the contact patch as the wheel moves along the rail. However, theoretical analysis [35] shows that when the diameter of the circular track is 2 meters, the difference in wheel-rail forces compared to straight-line motion is within 5%, which meets engineering requirements.

The test rig is designed with a dual-slip differential gear reducer. The vehicle speed motor, connected to the gear reducer via a clutch, drives the rotating arm to simulate train speeds. The wheel speed motor drives the wheels independently, allowing for slip ratio between wheels and rail of up to 100%. A torque sensor is installed on the wheel axle to measure torque and rotational speed, while a cylinder at the end of the rotating arm accurately simulates different axle loads. Compressed air is supplied through a collector ring and piping by an air compressor. Table 2 shows details of the parameters to be measured and the measurement methods.

Table 1. Technical indicators of the test rig.

| No. | Technical indicators | Parameter |
| --- | --- | --- |
| 1 | Scale ratio | 1:4 |
| 2 | Wheel diameter | 210 mm |
| 3 | Annular rail diameter | 2000 mm |
| 4 | Speed range | 0 ~ 160 km/h |
| 5 | Acceleration range | 0 ~ 2 m/s2 |
| 6 | Slip ratio range | 0 ~ 100% |
| 7 | Slip ratio change rate | 10%/s |
| 8 | Axle load range | 11 ~ 25 t |

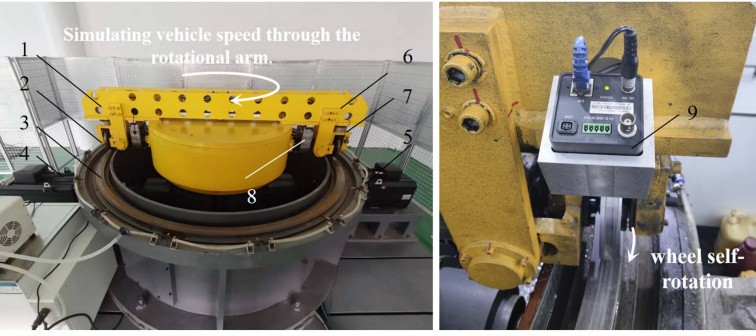

1 Rotational arm, 2 Wheels, 3 Track, 4 Vehicle speed motor, 5 Wheel speed motor, 6 Cylinder, 7 Pressure sensor, 8 Torque sensor，9 Thermal imager

**Fig 1. Circumferential test rig.**

**Table 2. Measurement methods and parameters.**

| Parameters | Measurement method |
|---|---|
| Adhesion coefficient $\mu$ | Calculated from the torque and axle load |
| Slip ratio $s$ | Calculate based on wheel speed and vehicle speed |
| Vehicle speed $v_v$ | Obtained through the vehicle motor |
| Wheel speed $v_w$ | Obtained through a torque meter |
| Axle load N | Obtained through pressure sensors |
| Tread temperature | Obtained through thermal imager |
| Acceleration | Obtained through acceleration sensors |
| Initial oil film thickness | Measure with a hexagonal comb gauge |
| Rail surface roughness | Obtained through a roughness measuring instrument |

By collecting data of vehicle speed $v_v$, wheel speed $v_w$, normal force between the wheels and rails N (axle load), and torque M from the torque sensor, the adhesion coefficient μ and slip ratio s between the wheels and rails are calculated in real-time during the experiment.

$$\mu = \frac{M + J\dot{\omega}}{NR} \tag{1}$$

$$s = \frac{v_v + v_w}{v_v} \tag{2}$$

In the Eqs. (1) and (2), M represents the torque measured by the torque sensor, J denotes the moment of inertia of the wheel axle, $\dot{\omega}$ is the angular acceleration of the wheel, and R is the radius of the wheel. Using this equation, a complete wheel-rail adhesion coefficient-slip ratio curve can be constructed.

## 2.2. Experimental procedure

Adhesion tests on oil condition were conducted using the circumferential test rig. In adhesion testing, to minimize unnecessary impacts and fluctuations between the wheel and rail and obtain higher-quality data, the target slip ratio is typically set as a smooth curve, such as the sine wave used in reference [8–13]. Excess oil is applied to the rail surface in this study to

ensure adequate lubrication at the wheel-rail interface, thereby maintaining consistency in the initial rail surface conditions. The experimental procedure is as follows:

(1) Prepare the Low-Adhesion Rail Surface: Use a brush to evenly apply 40 ml of oil (type 75W-90) onto the surface of the annular rail surface, ensuring sufficient lubrication of the wheel rail contact interface.

(2) Set the Required Axle Load: Adjust the axle load according to the current experimental conditions.

(3) Conduct adhesion test: Accelerate the vehicle and wheel to the specified speeds, set the wheel slip mode to simulate braking conditions, and record the speed and torque data during this process.

(4) Repeat the Experiment: Perform the test at least three times consecutively.

(5) Clean and Reapply Oil: After cleaning the surface oil contamination with a detergent, reapply 40ml of oil and conduct the next test.

(6) Complete the Experiment.

Through the above process, the adhesion coefficient slip ratio characteristics of the wheel rail under oil conditions can be obtained. The low adhesion track surface constructed is shown in Fig 2.

## 3. Experimental results

### 3.1. Adhesion characteristics under oil conditions

Existing literature indicates that under oil conditions, the wheel-rail adhesion coefficient-slip ratio curve typically shows only a single peak. However, most studies focus on slip ratio within the 0–10% range and lack exploration of adhesion characteristics under higher slip levels. To address this gap, this study conducted adhesion tests at higher slip levels to investigate whether low-adhesion oil-contaminated surfaces could be disrupted by wheel sliding. Fig 3 presents test result for an oil condition at 40 km/h and 11t axle load, with slip ratio of 20%, 40%, and 60%. As shown in Fig 3, under these low-speed and low-axle-load conditions, slip ratio of 20%, 40%, and even up to 60% did not disrupt the oil-contaminated rail surface, confirming the single-peak adhesion characteristic previously reported in the literature. In addition, it can be seen from Fig 3 that there is a slight difference in the peak point under different maximum slip ratio conditions, increasing from about 0.025 for a 20% slip ratio to about 0.05 for a 60% slip ratio. The author believes that the reason for this increase is due

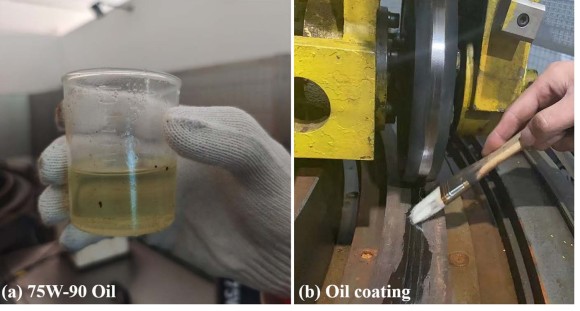

**Fig 2. Distribution of oil on the track surface.**

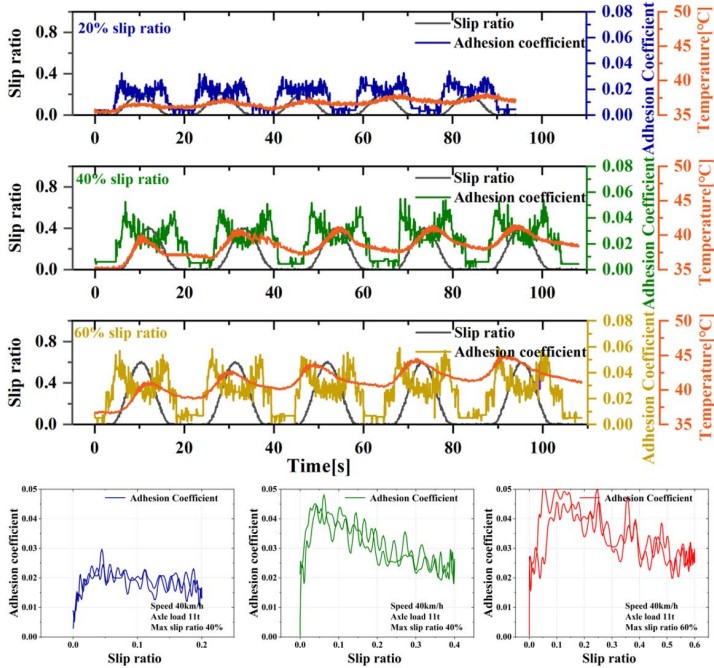

**Fig 3. Wheel-rail adhesion characteristics under oil conditions.**

to the heating of the oil medium under the action of large sliding, resulting in a decrease in viscosity.

As the slip ratio increases to 80%, the oil-contaminated wheel-rail surface can also exhibit an adhesion improvement phenomenon, as shown in Fig 4, which presents results from five consecutive tests. Initially, during the first test, the adhesion coefficient between the wheels and rails remains relatively low. However, with each subsequent test, the continuous wheel sliding leads to disruption of the oil-contaminated interface, resulting in a significant secondary increase in the adhesion coefficient. Notably, after the third wheel sliding test, the maximum adhesion coefficient on the wheel-rail surface rises to approximately 0.25, and each subsequent sliding test further improves adhesion significantly. From the adhesion coefficient-slip ratio curve, it is evident that in the initial test, the adhesion curves during increasing (braking) and decreasing (re-adhesion) slip ratio are nearly identical. However, in the later tests, the adhesion coefficient during the re-adhesion phase is significantly higher than during the braking phase, causing the adhesion coefficient-slip ratio curve to exhibit a ring-shaped pattern. The author attributes the oil-contaminated interface damage to the cumulative frictional heat generated by the mutual friction between the wheels and rails. During the initial slip, the accumulated energy is insufficient to disrupt the oil-contaminated surface. As the number of slidings increases, the frictional heat accumulated on the wheel surface exceeds the interface's threshold, making subsequent low-adhesion interface damage increasingly. The tests under extreme slip conditions indicate that oil-contaminated rail surfaces are not immune to disruption by wheel sliding but require more severe conditions compared to water.

## 3.2. The influence of speed on the oil condition

To investigate the impact of train speed on adhesion improvement under oil conditions, this study analyzed the results across different speeds. As illustrated in Fig 5, at a low speed of

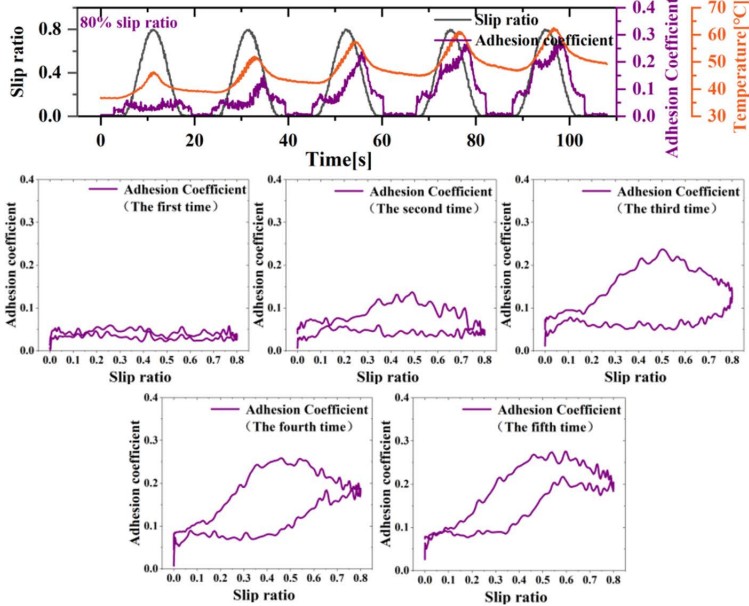

**Fig 4. Disruption of adhesion conditions at the oil-contaminated interface.**

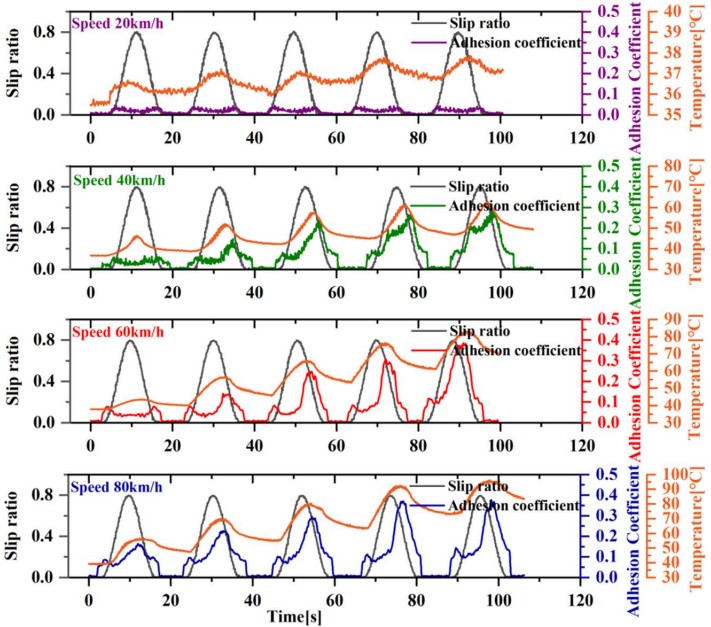

**Fig 5. The effect of different speeds on adhesion.**

20 km/h, even with a slip ratio of 80%, the adhesion coefficient continues to display a single peak. This suggests that the disruption of the oil-contaminated surface is not solely dependent on the slip ratio. It was observed that during each sliding, the temperature of the wheel surface increased slightly, by only 2–3 °C, which is not a substantial change. In contrast, at higher speeds, such as 60 km/h and 80 km/h, the trend of increasing adhesion coefficient is

more pronounced in comparison to the 40 km/h condition. It is noteworthy that at a speed of 80 km/h, the initial sliding test already demonstrates an increase in adhesion. This indicates that higher vehicle speeds result in a more pronounced deterioration of the oil-contaminated interface, despite the maintenance of the same slip ratio conditions. This may be attributed to the intensified friction between the wheels and rails at higher speeds, which causes a rapid rise in temperature until the oil medium reaches its thermal failure threshold, resulting in lubrication failure. However, at lower speeds or lower slip ratio conditions, the energy generated by wheel-rail friction is insufficient to disrupt the oil film. Consequently, the load between the wheels and rails is entirely supported by the oil film, and the adhesion coefficient reflects the shear characteristics of the oil film at this time.

The experiments revealed that in certain conditions, the oil film does not necessarily become damaged simply when the speed differential between the wheels and rails reaches a specific level during a single sliding test. Instead, it appears that the temperature of the wheels during multiple slidings has a cumulative effect. Once the accumulated temperature or energy exceeds a certain critical point, the oil film is disrupted, leading to a noticeable increase in the adhesion coefficient. However, under lower speed conditions, although multiple wheel slidings can accumulate thermal energy, the time interval between consecutive wheel passes over the same rail surface is relatively large. This allows sufficient time for heat dissipation, enabling a balance between thermal accumulation and dissipation. Consequently, the critical point for oil film failure is not reached, preventing significant changes in the adhesion coefficient.

### 3.3. The influence of axle load on the oil condition

Axle load is one of the significant factors affecting the adhesion coefficient, though its impact is typically less dramatic compared to speed. In the adhesion improvement process induced by wheel sliding studied in this paper, varying axle loads lead to differences in slip energy, making the effect of axle load also significant. As shown in Fig 6, which presents the adhesion variations under conditions of 40 km/h speed and axle loads ranging from 11 to 17t, the adhesion coefficient does not show an increase during the first sliding test under lower axle loads of 11t and 13t. In contrast, higher axle loads of 15 tons and 17 tons lead to surface disruption of the oil film even during the first sliding test. Furthermore, the oil-contaminated surface is more prone to be damaged under higher axle loads and shows a higher degree of adhesion increase. This is because higher axle loads result in greater energy during the same slip ratio, causing the oil to heat up more quickly and reach thermal failure sooner. However, compared to the effect of speed, the influence of axle load variation is less pronounced.

### 3.4. Adhesion characteristics under dry conditions

To provide a reference for low adhesion conditions between wheel and rail, this paper presents experimental research on wheel-rail adhesion under clean and dry conditions. By setting a target slip ratio as a sinusoidal curve, the variation of the wheel-rail adhesion coefficient within the 0–30% slip ratio range was measured. As shown in Fig 7, the wheel-rail adhesion characteristics under dry rail conditions can be described as follows: with the increase in slip ratio, the wheel-rail adhesion coefficient gradually rises from 0 to a certain saturation value and then decreases. This saturation value is approximately between 0.4 and 0.55 and decreases with increasing speed levels. The adhesion coefficient between the wheel and rail decreases as speed increases, a phenomenon that aligns with existing experimental results [9–11]. This effect may be attributed to the increased vibrations between the wheel and rail at higher speeds. The results of multiple experiments under dry rail conditions are consistently similar,

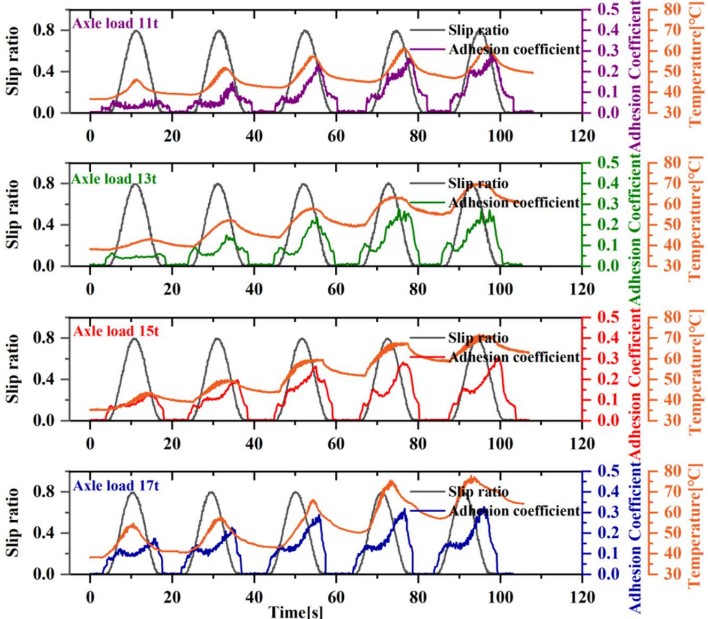

**Fig 6. The effect of different axle load on adhesion.**

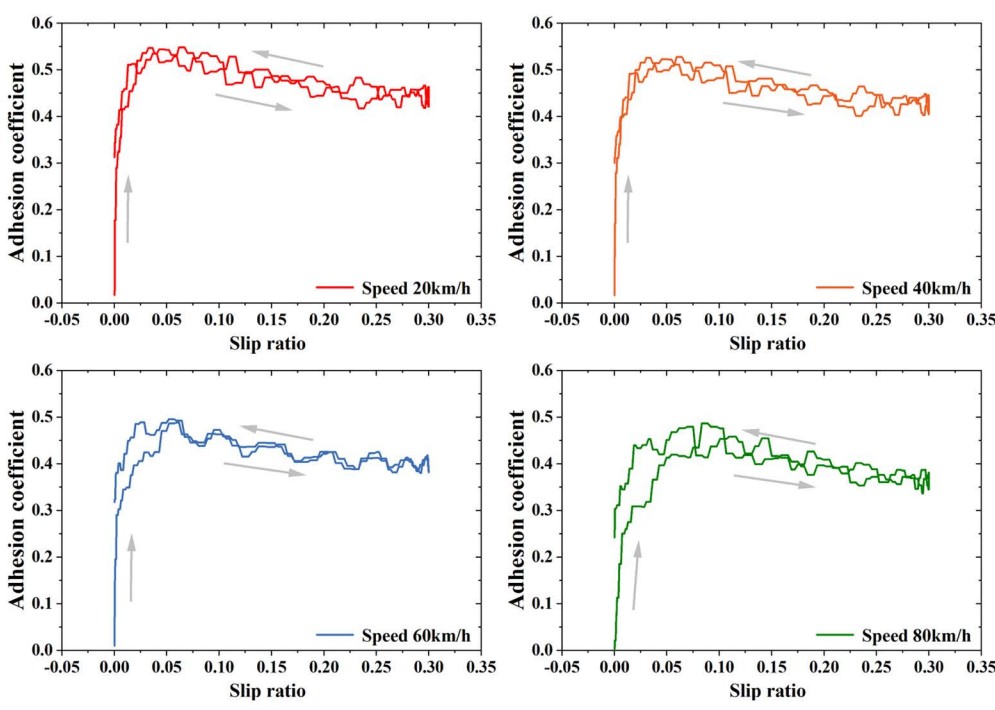

**Fig 7. Wheel-rail adhesion under dry conditions.**

and no improvement in adhesion is observed due to sliding between the wheels and rails. The adhesion coefficient of a dry rail surface can be understood as the upper limit of adhesion that can be achieved after the deterioration caused by low adhesion conditions due to oil contamination.

## 4. Discussion

### 4.1. Analysis of adhesion change mechanism

The experimental results under oil conditions in this study significantly differ from most traditional wheel-rail adhesion studies. In previous literature, the relationship between the adhesion coefficient and slip ratio typically shows a declining trend after reaching a saturation point, and the adhesion coefficient-slip ratio curve for oil-contaminated wheel-rail interfaces is a single-peaked curve. This is because past studies were conducted with lower slip levels, where the oil film on the wheel-rail surface was not disrupted by the sliding. Consequently, the contact conditions and friction state between the wheel and rail remained relatively constant. However, in this study, due to the severe frictional effects caused by high slip ratio in the oil conditions, the oil between the wheel and rail underwent lubrication failure. This transition caused the wheel-rail adhesion coefficient to move from a low-adhesion state to a higher adhesion state, ultimately approaching the dry rail surface condition, as shown in Fig 8.

Unlike the water conditions reported in the literature, a significant improvement in adhesion is observed at a slip ratio below approximately 30%, and the repeatability of the results is good throughout the continuous tests [7,8]. Under the oil condition, due to their higher viscosity compared to water, there is greater uncertainty and requiring multiple slidings to cause a secondary improvement in adhesion. Furthermore, the experimental results in this study show that under low-speed or small sliding conditions, even with repeated sliding, the oil-contaminated interface between the wheel and rail remains relatively constant and does not exhibit significant failure. However, once the lubricated interface between the wheel and rail is damaged for the first time, subsequent slips consistently lead to a secondary increase in the adhesion coefficient. This indicates that there is a specific condition for the lubrication failure of the oil interface, and reaching this condition can cause the low-adhesion interface to be disrupted. Most literature [12,13,18–20] on adhesion improvement under water condition suggests that the cumulative friction work or instantaneous friction power between the wheel and rail is a significant cause of adhesion improvement in water condition. During the experiments, the author observed through the thermal imager of the wheel-rail interface that when adhesion improvement occurs, the temperature of the wheel tread suddenly increases, and the wheel surface emits what appears to be oil fumes similar to that produced under high-temperature conditions, as shown in Fig 9. The temperature is taken from the average temperature value of the wheel surface. Therefore, the author concludes that the thermal lubrication failure of the oil interface between the wheel and rail is the fundamental reason for

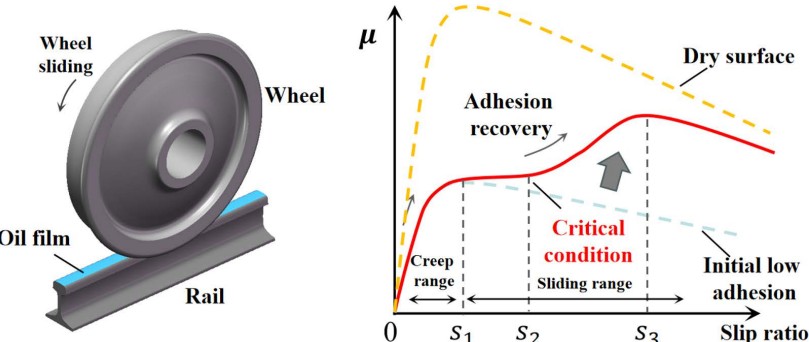

**High temperature caused by wheel sliding leads to failure of oil film lubrication**

**Fig 8. Oil lubrication failure on the wheel rail surface.**

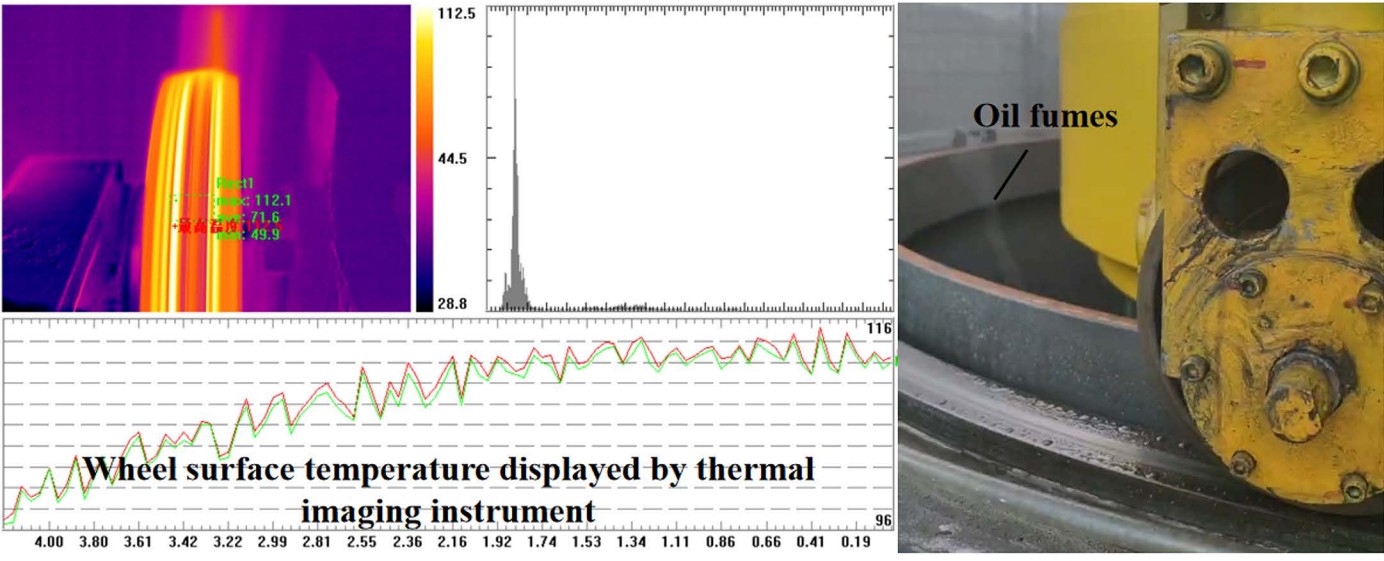

**Fig 9. Wheel-rail contact temperature in the test.**

the adhesion improvement. Additionally, intense friction between the wheel and rail not only accumulates temperature due to heating but also results in temperature reduction due to convective heat dissipation between the wheel, rail, and the atmosphere. This means that during wheel sliding, both heating from friction and cooling from heat dissipation occur. When the temperature increase significantly exceeds the temperature decrease, the temperature within the contact patch can quickly accumulate, reaching the critical temperature of the oil film and leading to its lubrication failure.

## 4.2. The impact of temperature

Based on the result in Section 3 and the analysis in Section 4.1, the authors believe that thermal lubrication failure of the oil film due to temperature rise is the fundamental cause of the observed adhesion improvement. Therefore, the paper analyzes the critical temperature at which the adhesion coefficient-slip ratio curve first shows an increase. Fig 10 below illustrates the temperature distribution histogram under different conditions. As shown in the Fig 10, the 75W-90 lubricant oil used in the experiments exhibits thermal failure at around 50°C under friction between the wheel and rail. This temperature range is similar to the critical temperature of some saturated fatty acids [36]. When the temperature between the wheel and rail reaches this critical point, the adhesion characteristic curve shows a distinct secondary increase, indicating that the oil medium interface has reached the failure condition.

Due to the instantaneous flash temperature within the wheel-rail contact patch, which is difficult to accurately capture with thermal imaging cameras or thermocouples, the critical failure temperature obtained in this study may be lower than the actual value. However, this does not affect our conclusion that the critical thermal failure temperature of the lubricant is indeed the failure condition for the oil-contaminated interface.

## 5. Conclusions

Based on the circumferential rail-wheel test rig, this study conducted adhesion characteristic tests under oil-contaminated conditions with a slip ratio up to 80%. The experiments included

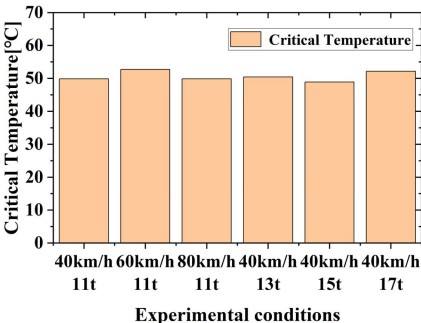

**Fig 10. Critical temperature for the tested conditions.**

various conditions such as different slip ratio, speeds, and axle loads. The study observed an improvement in adhesion under oil conditions and discussed the reasons behind this phenomenon. The main conclusions are as follows:

(1) Similar to water conditions, high slip ratio can also disrupt the low-adhesion interface caused by oil, leading to an improvement in wheel-rail adhesion characteristics. However, the conditions required to disrupt the low-adhesion interface with oil-contaminated are more stringent than that of water.

(2) Speed, axle load, and slip ratio all affect the disruption of the oil-contaminated interface. A higher slip ratio is a necessary condition for disrupting the low-adhesion interface. Under the same slip ratio conditions, higher speed and axle load can make it easier for the oil-contaminated interface to experience adhesion improvement.

(3) The high temperatures generated by significant slip between the wheel and rail are a key factor in the lubrication failure of the oil film. The critical failure temperature of the oil medium is the condition that leads to the disruption of the oil-contaminated interface.

## Nomenclature

$\mu$: Adhesion coefficient

$s$: Slip ratio

$v_v$: Vehicle speed

$v_w$: Wheel speed

$N$: Axle load

$M$: Torque measured by the torque sensor.

## Author contributions

**Writing – original draft:** Zhou Gaowei, Zhou Jiajun, Tian Chun, Fei Gao.

**Writing – review & editing:** Zhou Gaowei, Zhou Jiajun, Tian Chun, Fei Gao.

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
