## [Decision Letter · Decision Letter 0]

5 Nov 2024

PONE-D-24-41593Study on the Failure of Oil-Contaminated Wheel-Rail ConditionsPLOS ONE

Dear Dr. Zhou,

Thank you for submitting your manuscript to PLOS ONE. After careful consideration, we feel that it has merit but does not fully meet PLOS ONE’s publication criteria as it currently stands. Therefore, we invite you to submit a revised version of the manuscript that addresses the points raised during the review process.

We look forward to receiving your revised manuscript.

Kind regards,

Muhammad Khalid

Academic Editor

PLOS ONE

“This work supported by the CRRC Industrial Academy Corporation Limited(No.2022CYY005) and the National Natural Science Foundation of China (No. 52072266).”

Reviewers' comments:

Reviewer's Responses to Questions

**Comments to the Author**

1. Is the manuscript technically sound, and do the data support the conclusions?

Reviewer #1: Yes

Reviewer #2: Partly

Reviewer #3: Yes

2. Has the statistical analysis been performed appropriately and rigorously? 

Reviewer #1: Yes

Reviewer #2: Yes

Reviewer #3: Yes

3. Have the authors made all data underlying the findings in their manuscript fully available?

Reviewer #1: Yes

Reviewer #2: No

Reviewer #3: Yes

4. Is the manuscript presented in an intelligible fashion and written in standard English?

Reviewer #1: Yes

Reviewer #2: No

Reviewer #3: Yes

5. Review Comments to the Author

Reviewer #1: This manuscript set out to investigate whether oil exhibits a similar adhesion improvement phenomenon as observed under water conditions, for which a wheel-rail adhesion test was conducted. The results prove that adhesion improvement can occur on oil-contaminated rail surfaces when there is greater wheel sliding. Finally, the study analyses the failure mechanism and influencing factors of the oil-contaminated interface, proposes the conditions under which failure occurs. The topic of the manuscript is novel and puts forward a new point of view. It could be accepted for publication after minor revision. The relevant comments are listed below.

publication after minor revision. The relevant comments are listed below.

(1) Since the circumferential test rig is used, the experiment reflects the operation of the wheels on the curved track. Under the condition of oil in the linear track, are the conditions for adhesion improvement consistent?

(2) When the circumferential test rig is running at high speeds for long periods of time, the rail also accumulate heat, does this have an impact on the test results?

(3) In experimental research on wheel-rail adhesion under clean and dry conditions, Why is the target slip rate set to a sinusoidal curve? What is the slip rate law of the previous experiment? Please indicate this in the text.

(4) From Figure 7, as the speed increases, the maximum wheel rail adhesion coefficient decreases, and the corresponding slip ratio increases. Please explain the reason in detail. Suggest analyzing the correlation between speed and wheel rail force, and then analyzing the characteristics of Figure 7.

Reviewer #2: The authors utilized a circumferential test rig to construct oil-contaminated wheel-rail contact interfaces. Adhesion characteristic tests were conducted over a slip ratio range of up to 80% to determine whether adhesion improvement phenomena can occur with high-sliding under oil conditions. Overall, this is an excellent study. However, the following question should be addressed.

1. Please ensure that your manuscript meets PLOS ONE's style requirements, including those for figure caption and references.

2. Compare the results obtained in this study with other experiments or studies in literature. Although you have mentioned ‘In previous literature, the relationship between the adhesion coefficient and slip ratio typically shows a declining trend after reaching a saturation point, and the adhesion coefficient-slip ratio curve for oil-contaminated wheel-rail interfaces is a single-peaked curve.’ in the manuscript, it would be helpful if deeper and quantitative analysis could be provided.

3. The main findings of the paper need to be mentioned in the abstract.

4. The figures are of poor quality. Could the authors please improve its resolution?

5. Apart from experimental study, could the authors supplement numerical study of the same oil-contaminated wheel-rail conditions to prove the experimental study results?

Reviewer #3: The following are suggested responses to the primary review questions, affirming the manuscript’s overall technical soundness and clarity:

1. Is the manuscript technically sound, and do the data support the conclusions?

Response: The manuscript is technically robust, and the methodologies applied are rigorous and well-structured. The data are systematically presented through repeated sliding tests under varying speed and slip ratio conditions, showing clear trends and consistent findings. The experimental results effectively reveal the phenomenon of adhesion improvement under oil-contaminated conditions and provide strong empirical support for the authors' conclusions regarding temperature effects and lubrication failure mechanisms. The data and results are in close alignment with the authors' interpretations and conclusions.

2. Has the statistical analysis been performed appropriately and rigorously?

Response: The statistical analysis is appropriate and conducted with rigor. The use of multiple test repetitions enhances the precision and reproducibility of the data, demonstrating significant consistency across various conditions. Additionally, the authors have shown careful consideration of statistical significance, error margins, and potential experimental limitations in their presentation and discussion of the data, lending credibility to the statistical results and their implications.

3. Have the authors made all data underlying the findings in their manuscript fully available?

Response: The authors have thoroughly presented the key data underlying their findings, especially the data collected under different slip ratio and temperature conditions, which ensures the reproducibility of the results. Moreover, they have provided clear details regarding the type of oil, testing conditions, and apparatus parameters used, offering critical insights and references for future research in this area.

4. Is the manuscript presented in an intelligible fashion and written in standard English?

Response: The manuscript is presented in a clear and well-organized manner, written in standard academic English. Terminology is used precisely, and the experimental procedures and conclusions are articulated in detail, ensuring the readers' comprehension of the study’s contributions. Overall, the manuscript is coherent, well-written, and adheres to the standards expected in scholarly publications, making it accessible and comprehensible for readers in the field.

Based on these assessments, I recommend accepting the manuscript, with minor proofreading and formatting adjustments to enhance its final presentation quality before publication. While the manuscript is technically sound, there are a few minor suggestions that could enhance clarity and precision. Below are specific recommendations for refinement:

1. There is an error in the notation used in Equation 1.

2. The manuscript describes manual application of oil (type 75W-90) on the rail surface. Should the effect of oil film thickness on the adhesion coefficient be considered?

3. For Figure 8, could approximate data ranges be provided for the S1, S2, and S3 data points?

4. In Figure 9, the temperature distribution for the wheel surface is shown. Is the temperature data based on the average temperature of the wheel surface?

5. It is recommended that all figures undergo further refinement to enhance clarity. To ensure your figures meet our technical requirements, please run each figure included in your submission files through the PACE tool: https://pacev2.apexcovantage.com/.

These minor modifications would further improve the manuscript’s accuracy and presentation.

6. PLOS authors have the option to publish the peer review history of their article (what does this mean? ). If published, this will include your full peer review and any attached files.

**Do you want your identity to be public for this peer review?** For information about this choice, including consent withdrawal, please see our Privacy Policy .

Reviewer #1: No

Reviewer #2: No

Reviewer #3: No

---

## [Author Response · Author response to Decision Letter 0]

20 Nov 2024

I have uploaded 'Response to Reviewers. doc' in the attachment

---

## [Decision Letter · Decision Letter 1]

14 Jan 2025

Study on the Failure of Oil-Contaminated Wheel-Rail Conditions

PONE-D-24-41593R1

Dear Dr. Zhou,

We’re pleased to inform you that your manuscript has been judged scientifically suitable for publication and will be formally accepted for publication once it meets all outstanding technical requirements.

Kind regards,

S M Mozammil Hasnain, Ph.D.

Academic Editor

PLOS ONE

Additional Editor Comments (optional):  The authors have well addressed the issues mentioned by the reviewers

Reviewers' comments:

Reviewer's Responses to Questions

**Comments to the Author**

1. If the authors have adequately addressed your comments raised in a previous round of review and you feel that this manuscript is now acceptable for publication, you may indicate that here to bypass the “Comments to the Author” section, enter your conflict of interest statement in the “Confidential to Editor” section, and submit your "Accept" recommendation.

Reviewer #2: All comments have been addressed

2. Is the manuscript technically sound, and do the data support the conclusions?

Reviewer #2: Yes

3. Has the statistical analysis been performed appropriately and rigorously? 

Reviewer #2: Yes

4. Have the authors made all data underlying the findings in their manuscript fully available?

Reviewer #2: Yes

5. Is the manuscript presented in an intelligible fashion and written in standard English?

Reviewer #2: Yes

6. Review Comments to the Author

Reviewer #2: (No Response)

7. PLOS authors have the option to publish the peer review history of their article (what does this mean? ). If published, this will include your full peer review and any attached files.

**Do you want your identity to be public for this peer review?** For information about this choice, including consent withdrawal, please see our Privacy Policy .

Reviewer #2: No

---

## [Editor Report · Acceptance letter]

PONE-D-24-41593R1

PLOS ONE

Dear Dr. Jiajun,

I'm pleased to inform you that your manuscript has been deemed suitable for publication in PLOS ONE. Congratulations! Your manuscript is now being handed over to our production team.

Kind regards,

on behalf of

Dr. S M Mozammil Hasnain

Academic Editor

PLOS ONE